# A High Prevalence of Vitamin D Deficiency Observed in an Irish South East Asian Population: A Cross-Sectional Observation Study

**DOI:** 10.3390/nu12123674

**Published:** 2020-11-28

**Authors:** Eamon Laird, James Bernard Walsh, Susan Lanham-New, Maria O’Sullivan, Rose Anne Kenny, Helena Scully, Vivion Crowley, Martin Healy

**Affiliations:** 1School of Medicine, Trinity College Dublin, Dublin 2, Ireland; mosulli5@tcd.ie (M.O.); rkenny@tcd.ie (R.A.K.); 2Mercer’s Institute for Successful Ageing, St. James’s Hospital, Dublin 8, Ireland; jbwalsh@tcd.ie (J.B.W.); helenascully@hotmail.com (H.S.); 3Department of Nutritional Science, University of Surrey, Surrey GU2 7YW, UK; s.lanham-new@surrey.ac.uk; 4Department of Biochemistry, Central Pathology, St. James’s Hospital, Dublin 8, Ireland; vcrowley@stjames.ie (V.C.); healymartinj@gmail.com (M.H.)

**Keywords:** vitamin D, population, Asian, minority, ethnic, health

## Abstract

At northern latitudes, non-ethnic population groups can be at an increased risk of vitamin D deficiency (defined as a 25-hydroxyvitamin D [25(OH)D] status ≤30 nmol/L). The vitamin D status of ethnic minority groups has been examined both in UK and European populations, but not in the Irish context. The aim of this study is to assess the vitamin D status from a selection of the Dublin population of South East Asian descent. A search was conducted, using the laboratory information system of St James’s Hospital, Dublin, for vitamin D requests by General practitioners. From 2013 to 2016, 186 participants were identified and 25(OH)D analysis was quantified using liquid chromatography-tandem mass spectrometry (LC-MS-MS). Overall, the median age was 32 years, 51% were male, and the 25(OH)D concentration ranged from 10 to 154 nmol/L. In total, 66.7% of the total sample were vitamin D deficient and 6.7% had a 25(OH)D status greater than 50 nmol/L (the 25(OH)D concentration defined by the EU as ‘sufficient’). Females had a significantly higher 25(OH)D concentration than males (25.0 vs. 18.0 nmol/L; *p* = 0.001) but both groups had a significant proportion with deficient status (56% and 76.8%, respectively). Seasonal variation of 25(OH)D was not evident while high rates of deficiency were also observed in those aged <18 years and >50 years. Given the importance of vitamin D for health, this sub-population could be at a significantly increased risk of rickets, impaired bone metabolism, and osteoporosis. In addition, vitamin D deficiency has been associated with several non-bone related conditions, including cardiovascular disease and diabetes. Currently, there is no unique vitamin D intake or vitamin D status maintenance guidelines recommended for adults of non-Irish descent; this needs to be considered by the relevant public health bodies in Ireland.

## 1. Introduction

Vitamin D is a seco-steroid hormone with deficiency (<30 nmol/L) associated with impaired bone metabolism and increased risk of osteoporosis [1]. It has also been associated with extra-skeletal health outcomes such as cardiovascular disease (CVD), cancers, diabetes, and inflammation in many observational and prospective studies [2,3,4]. The main source of vitamin D is exposure to solar ultraviolet-B radiation at the correct wavelength though factors influencing this process, including latitude, seasonality, sunscreen use, ethnicity, clothing, and long periods indoors [5]. Due to the seasonality of vitamin D synthesis at Northern latitudes, there is a heavy reliance on dietary intakes during the winter period to maintain adequate circulating concentrations [6]. Unfortunately, foods that are rich dietary sources of vitamin D are infrequently consumed and many food products are not fortified with vitamin D [6]. Thus, significant rates of vitamin D deficiency and insufficiency have been reported in countries such as Ireland, the UK, and several other European states [7]. For instance, recent data from the Irish Longitudinal Study on Aging (TILDA) reported one in eight older Caucasian Irish adults were vitamin D deficient, which increased to one in four during the winter period [8].

Recent studies have shown that the vitamin D status of immigrant populations is significantly poorer when compared to the indigenous population of countries investigated [9]. This is particularly true of the migrant Asian population where a large number of investigations in the UK have demonstrated a high prevalence of hypovitaminosis D exists in this group [10,11,12,13]. For instance, in 6433 South Asians from the UK Biobank, 92% had blood vitamin D levels <50 nmol/L [14]. To date, however, there has been no estimation of vitamin D status in the Asian migrant population in Ireland, a group comprising approximately 79,000 (1.7%) of the total population [15].

The aims of this cross-sectional observational study, therefore, are to assess the vitamin D status for a selection of the Dublin (53.3° N; capital city) population of South East Asian descent and to provide pilot data for this population. This may allow a future intervention strategy to be considered.

## 2. Materials and Methods

### 2.1. Study Design

A search was conducted using the Biochemistry Department laboratory information system (iSOFT Telepath^®^) of St James’s Hospital, Dublin, Ireland (53.3° N) for vitamin D requests by general practitioners (GPs) known to take patients self-identified as Asian (non-Chinese). From 2013 to 2016, 186 participants were found. After identification, all participants were given an anonymized code and recognition materials were removed from the analysis database. The exclusion criteria included those who were non-Asian, aged <2 years, missing demographic data (such as age or sex), or were residing in nursing home locations. Seasons were defined as Winter: (December–February), Spring: (March–May), Summer: (June–August), and Autumn: (September–November). Duplications within a specified season were averaged, with the duplicate being excluded. Seasons were collated and, given the latitude of Ireland, grouped in either low vitamin D synthesis period (October to March) or high vitamin D synthesis period (April to September). The joint research ethics committee at St James’s Hospital/Tallaght University Hospital (SJH/TUH) granted ethical approval for this study (Ref: 5475), which was conducted according to the guidelines laid down in the Declaration of Helsinki 1964.

### 2.2. Laboratory Analysis

Samples for vitamin D analysis included total 25-hydroxy-vitamin D (25(OH)D) (D2 and D3) concentrations, which were quantified by a validated method (Chromsystems Instruments and Chemicals GmbH; MassChrom 25-OH-Vitamin D3/D2) using liquid chromatography-tandem mass spectrometry (LC-MS-MS) (API 4000; AB SCIEX, Framingham, MA, USA) and analyzed in the Biochemistry Department of St James’s Hospital (accredited to ISO 15189). The quality and accuracy of the method was monitored by the use of internal quality controls, participation in the Vitamin D External Quality Assessment Scheme (DEQAS), and the use of the National Institute of Standards and Technology (NIST) 972 vitamin D standard reference material. The respective inter- and intra-assay coefficients of variation were 5.7% and 4.5%. For this study, vitamin D ‘sufficiency’ was defined as a serum 25(OH)D concentration ≥50 nmol/L, vitamin D ‘insufficiency’ as 30 to 49.9 nmol/L, and risk of vitamin D deficiency as <30 nmol/L [1]. High vitamin D status was defined as >125 nmol/L [1]. Intact parathyroid hormone (PTH) was measured at St. James’s Hospital, Dublin using an electrochemiluminescence immunoassay (ECLIA) (Modular E170, Roche Diagnostics, Burgess Hill, UK) with an inter-assay CV of <2.9% and an assay measurement range of 1.2 to 5000 pg/mL. Calcium was assayed on a Roche c701 chemistry analyzer (Roche Diagnostics, Burgess Hill, UK) using a proprietary calcium kit (Calcium Gen 2). The assay measurement range was 0.20 to 7.5 nmol/L with an interassay precision of <2.6%.

### 2.3. Statistical Analysis

Statistical analyses were carried out using the SPSS, version 24.0 (IBM Corp., Armonk, NY, USA, 2019). Data were checked for normality by the Kolmogorov-Smirnov test and Q-Q plots and, where appropriate, data were log-transformed. Data within the tables are expressed as geometric-means with standard deviation (SD). Where appropriate, an independent Student’s *t*-test, one-way ANOVA, or, for categorical variables, chi-square analysis or Fisher’s (where *n* was <5) were applied to determine statistical significance. Statistical significance was accepted at a *p* value of <0.05).

## 3. Results

The population characteristics are displayed in Table 1 (*n* 186). Overall, the median age was 32 years, 51% were male, and <5% were aged <18 or >50 years. There was no significant difference in the mean age between genders, though females had a slightly higher proportion aged 18 to 50 years (*p* = 0.047). The sample contained significantly less females than males from the winter period (19.8% vs. 44.2%; *p* = 0.001) and a slightly higher proportion of females during autumn (37.3% vs. 21.1%; *p* = 0.011). However, females had a significantly higher PTH (*p* = 0.044) and lower calcium (*p* = 0.004) concentration in comparison to males. The total sample 25(OH)D concentration ranged from 10 to 154 nmol/L with females having a higher median 25(OH)D compared to males (25.0 vs. 18.0 nmol/L; *p* = 0.004) but both groups still fell below the risk of deficiency cut-point of 30 nmol/L. The average 25(OH)D concentration for those aged <18 years was 20.0 nmol/L (14.0 to 37.0 nmol/L) while for those aged >50 years it was 26.0 nmol/L (14.5 to 50.5 nmol/L). Only one participant (0.5% of the sample) had a 25(OH)D concentration >125 nmol/L.

In total, 66.7% of the total sample were vitamin D deficient; males and younger adults (<18 years) had higher deficiency rates in comparison to females and older adults (>50 years). When the seasons were grouped into the ‘low’ and ‘high’ vitamin D synthesis periods, there was no significant difference in 25(OH)D concentration between the two periods (20.0 vs. 23.0 nmol/L, respectively; *p* = 0.223). Males had a higher percentage of deficiency compared to females in the low synthesis period (79.3 vs. 51.5%; *p* = 0.009) and a lower percentage with sufficient status (5.2 vs. 27.3%; *p* = 0.004) (Figure 1). There was no gender difference when examined by the high synthesis period (Figure 2). All age groups had ≥50% risk of 25(OH)D deficiency regardless of the synthesis period sampled (Figure 3 and Figure 4).

In relation to biochemical markers of bone health (Table 2), concentrations of PTH were significantly higher in participants who were 25(OH)D deficient compared to sufficient (45.5 vs. 36.1 pg/mL).

## 4. Discussion

To our knowledge, this is the first study in Ireland to examine the vitamin D status of Irish South East Asians. Our findings reflect previous studies demonstrating poor vitamin D status among non-western immigrants living in European countries. In this cohort, over 66% were vitamin D deficient, which was more than five times the estimated deficiency rate for Caucasian Irish adults (13%). These high rates of deficiency were observed regardless of vitamin D synthesis period, gender, or age. In addition, participants with vitamin D deficiency had a significantly higher PTH concentration compared to those who were sufficient. Raised PTH reflects low or insufficient vitamin D status and the combination of both of these can be associated with high bone turnover, which may result in net bone resorption [1]. Extended periods of vitamin D deficiency have been associated with muscle weakness, bone demineralization resulting in rickets or osteomalacia, pain, fractures, and frailty.

This population group could also be at risk of other non-skeletal chronic diseases that are thought to be associated with vitamin D deficiency including type 2 diabetes and heart disease [2,3]. Several reports have shown that South East Asian populations have an increased incidence prevalence of these conditions [16,17]. In the current context of COVID-19, research evidence [18] has shown that this group may also have poorer outcomes with some suggesting a possible mechanistic link with low vitamin D status given its role in supporting the immune system [4,19].

Our findings of high deficiency rates are similar to observations from the UK and Europe [9,10,11,12,13,14]. High rates of deficiency have been reported in South Asians arriving in Norway (75% deficient) [20] along with low 25(OH)D concentrations in Asian children residing in London [21] and an increased risk of osteomalacia in adult Asian groups also living in the UK [22]. These high deficiency rates are not just limited to the Asian ethnic group but appear to be common among most immigrant groups that travel from the lower latitudes to the Northern latitude countries [23,24,25]. A number of reasons for the high deficiency rates can be speculated. South East Asians typically have higher concentrations of skin melatonin resulting in a longer time period required to synthesize vitamin D at higher latitude countries such as Ireland [26]. This could further be compounded by the fact that in Ireland meaningful vitamin D synthesis can only occur from late March to early September while the temperate cloudy climate can further reduce synthesis [27]. Furthermore, the South East Asian population has a rich cultural history where reduced skin exposure to UV light may result in a further reduction of vitamin D synthesis [28]. However, in our study, men typically had higher levels of deficiency than women—this is a common observation that is being reported across many vitamin D studies. There is no firm hypothesis to account for this difference—it is possible the different fat mass and muscle composition may have an effect while differences in the metabolic utilization or tissue priority for vitamin D could be different in the sexes.

In terms of dietary intakes of vitamin D, scant information exists of the vitamin D dietary intakes of Irish Asians. The traditional South East Asian diet has been reported to be low in foods containing vitamin D [29]. Mandatory fortification of foods with vitamin D does not occur in Ireland and it is subject only to a voluntary ‘custom’ with few foods containing vitamin D apart from some milk and yoghurt products. Other dietary lifestyle factors may influence vitamin D including specific cultural practices such as Betel nut chewing which is common among Asian populations. This aggravates vitamin D deficiency by increasing 25-hydroxylase activity and decreasing 1,25-di-hydroxyvitamin D systemic concentrations [30]. There is no epidemiological information regarding Betel nut chewing among the Irish Asian population.

Recently, the Food Safety Authority of Ireland (FSAI) updated the national vitamin D intake guidelines for the Irish population as a whole. [31]. In the policy document, it was recommended that food (including supplements) needs to provide 10 µg of vitamin D every day for everyone aged ≥5 years. Few foods, however, contain the necessary vitamin D concentrations and those that do are infrequently consumed. This may be due to lack of availability or costs of the food products. There may also be cultural traditions regarding different food types including, in particular, vegan diets. Importantly in the current context, although other international policy documents address the vitamin D needs of specific population sub-sets, the FSAI does not do so. However, across the EU there is little harmonization of vitamin D supplementation and fortification policies and mandatory food fortification with vitamin D is rarely implemented. Furthermore, policy documents can include confusing and contradictory information, which makes it difficult to translate guidelines to specific settings [32]. In the UK, both the Scientific Advisory Committee on Nutrition (SACN) and the National Institute for Health and Care Excellence (NICE) have made recommendations for ‘at risk’ groups regarding supplementation with vitamin D [33,34]. These risk groups include those on vegetarian diets, those who have dark skin pigmentation, those who have limited exposure to UVB light because of cultural or lifestyle practices, and would include those in the Asian community. A ‘multi-agency’ approach is recommended to improve communication and education relating to vitamin D intake and to encourage behavioral involvement in the uptake of recommendations.

This study gives an overview of the vitamin D status of a sub-set of the Irish Asian population and is the first report to do so. It reviews both males and females across the age spectrum. Vitamin D analysis was assessed using LC-MS/MS which is the gold standard for vitamin D assessment. Limitations include the small sample size with no information on dietary vitamin D intakes, sun exposure, other demographics, or medication/supplement use as this information was not collected on the GP electronic sample system. The study was performed, however, to illustrate low vitamin D levels in this ethnic group with a view to designing follow-up studies and formulating an approach to assist in correcting the vitamin D deficit.

## 5. Conclusions

In conclusion, we observed high levels of vitamin D deficiency in a sample of South East Asian Irish. Their 25(OH)D concentration remained low throughout the year and was unaffected by seasonality. Vitamin D is universally accepted to have a critical role in normal bone metabolism. In addition, many studies have associated low vitamin D status with numerous chronic and systemic conditions. This population group is known to be at a higher risk for cardiovascular disease/diabetes and an association with chronically low vitamin D levels has been speculated. Its role in a normally functioning immune system has also been brought to the fore with the onset of COVID-19. It is critically important that the Government agencies tasked with developing policies for vitamin D requirements address sub-groups such as those described here who are at particular risk for deficiency.

## Figures and Tables

**Figure 1 nutrients-12-03674-f001:**
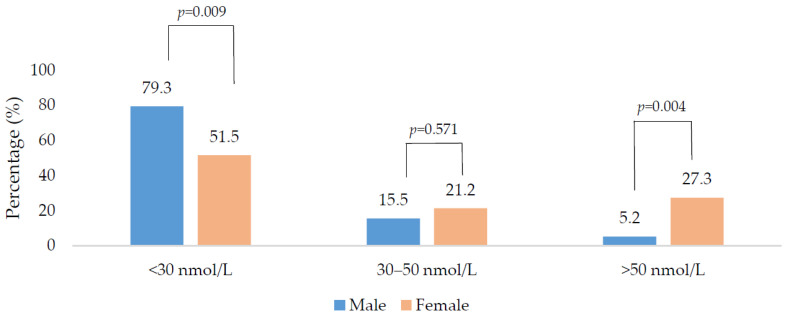
Vitamin D status of the population during the low vitamin D synthesis period (October to March) sub-divided by gender. *p*-values were derived from chi-square analysis.

**Figure 2 nutrients-12-03674-f002:**
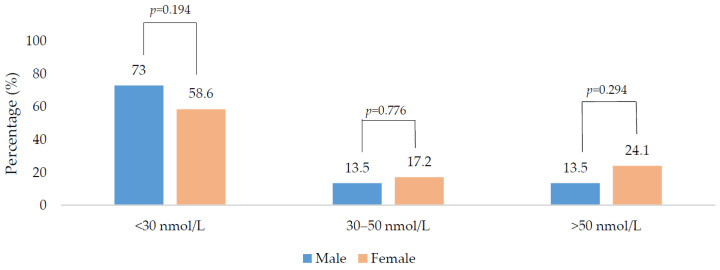
Vitamin D status of the population during the high vitamin D synthesis period (April to September) sub-divided by gender. *p*-values were derived from chi-square analysis.

**Figure 3 nutrients-12-03674-f003:**
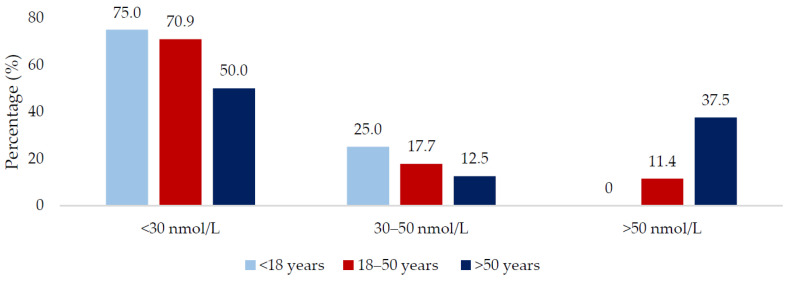
Vitamin D status of the population during the low vitamin D synthesis period (October to March) sub-divided by age group.

**Figure 4 nutrients-12-03674-f004:**
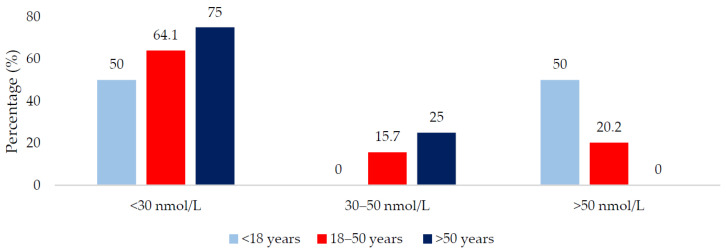
Vitamin D status of the population during the high vitamin D synthesis period (April to September) sub-divided by age group.

**Table 1 nutrients-12-03674-t001:** Population characteristics of the South East Asian participants ^1^.

	Male	Female	*p*-Value
	*n* 95	*n* 91	
Age (years)	31.0 (27.0, 36.0)	33.0 (26.0, 38.0)	0.178
<18 *n* (%)	2 (2.1)	4 (4.4)	0.437
18 to 50 *n* (%)	90 (94.7)	78 (85.7)	0.047
>50 *n* (%)	3 (3.2)	9 (9.9)	0.077
Season sampled *n* (%)			
Winter	42 (44.2)	18 (19.8)	0.001
Spring	17 (17.9)	16 (17.6)	0.956
Summer	16 (16.8)	23 (25.3)	0.207
Autumn	20 (21.1)	34 (37.3)	0.016
Bone biochemistry			
25(OH)D (nmol/L)	18.0 (27.0, 36.0)	25.0 (17.0, 30.0)	0.004
PTH (pg/mL)	41.6 (31.7, 53.8)	43.8 (35.1, 71.0)	0.044
Calcium (mmol/L)	2.35 (2.29, 2.41)	2.26 (2.22, 2.35)	<0.001

^1^ Values are displayed as *n* (%) or medians (25 to 75th percentile). Differences between genders for continuous variables were assessed using an independent *t*-Test while differences in categorical variables were assessed using chi-square analysis or Fisher’s where *n* was <5. Normal reference range for parathyroid hormone (PTH): 15 to 65 pg/mL; Normal reference range for calcium: 2.15 to 2.50 mmol/L.

**Table 2 nutrients-12-03674-t002:** Bone marker concentrations of the participants by 25(OH)D status ^1^.

	<30 nmol/L (Deficient)	30–50 nmol/L (Insufficient)	>50 nmol/L (Sufficient)
Total			
PTH (pg/mL)	45.5 (36.5, 65.1) ^a^	41.8 (28.1, 50.0) ^a,b^	36.1 (25.4, 43.8) ^b^
Calcium (mmol/L)	2.33 (2.25, 2.39)	2.32 (2.25, 2.38)	2.26 (2.24, 2.35)
Men			
PTH (pg/mL)	45.5 (33.1, 58.1) ^a^	37.1 (28.9, 46.1) ^a^	20.4 (16.6, 39.2) ^b^
Calcium (mmol/L)	2.35 (2.31, 2.41)	2.35 (2.30, 2.43)	2.26 (2.21, 2.37)
Women			
PTH (pg/mL)	45.8 (38.9, 89.0)	46.3 (25.6, 55.7)	39.3 (33.2, 48.1)
Calcium (mmol/L)	2.25 (2.20, 2.35)	2.26 (2.19, 2.37)	2.27 (2.24, 2.35)

^1^ Values are displayed as medians (25 to 7th percentile). Differences were assessed by One Way ANOVA with Bonferroni correction. Values in the same row with different superscript letters are significantly different, *p* < 0.05. Normal reference range for PTH: 15 to 65 pg/mL; Normal reference range for calcium: 2.15 to 2.50 mmol/L.

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
