# Peer review of "A High Prevalence of Vitamin D Deficiency Observed in an Irish South East Asian Population: A Cross-Sectional Observation Study"

_nutrients, 2020, doi:10.3390/nu12123674_

Round 1

Reviewer 1 Report

This study investigates the prevalence of 25OHD deficiency and insufficiency in an ethnic minority population of South East Asians living in Ireland. The study results of the study indicate a high prevalence of 25OHD deficiency/insufficiency, which is higher in men than in women. It also shows that there is no differences between the high vs. low synthesising months.

General comments:

Introduction:

Line 48 – 54: the authors should be more specific when describing the different ethnicity populations.

Line 55: “The aims of this study, therefore, were to assess the vitamin D status for a selection of the Dublin (53.3°N; Capitol City) population of South East Asian descent and to provide baseline pilot data for this population”. I suggest that the authors explain what they mean by “baseline pilot data”. Is this an intervention study and the authors are presenting the baseline results? Is this preliminary data from a larger study?

Methods:

The authors should follow the STROBE checklist for the description of this article (introduction, methods, results and discussion)

The authors have collected the necessary information to report 25OHD status; however, there is key information missing here. For example, confounding factors that may contribute to 25OHD deficiency and insufficiency apart from age and gender. For instance, ethnicity should be more specific, clinical history and any medications, socio-economic status, body mass index, vitamin D supplement usage, work and work patterns. Other information such as dietary intake (or cooking traditions), exposure to sunlight and time spent indoors, skin pigmentation and/or physical activity levels if possible.

The authors should explain the rationale for choosing age and gender as main independent variables and for stratification purposes.

Line 59 Study design: please state what type of study design is your study

Lines 75 – 90: please describe all the reference ranges used for your study including PTH. Did the authors collect phosphate? This is also an important marker of bone health.

Line 85: reference is missing

Line 88: reference is missing

Lines 91 – 97: Statistical analysis. I suggest you describe the rationale for using each test. For example, fisher exact test was used to establish associations between vitamin D status and gender.

Results:

As per STROBE, please add a flow diagram

Interquartile range (IQR) should accompanied median values. Please add this where relevant throughout the manuscript

All p values should be low case and italic

Line 99: indicate the total sample size

Line 110: “were vitamin D deficient, with 16.7% sufficient”. Please delete with.

Lines 119 – 121: the authors should stratify the data by age and look at whether there is any correlation between 25OHD and PTH in children

Line 122: table 1. Please make sure decimal places are the same in all figures.

Line 127: figure 1 and 2 could be joined into one for comparison reasons and to summarise more information into one figure.

Figures 1-4: please use the same format, add the n values and do not add values on top of columns.

Table 2: add reference values at the bottom for calcium and PTH

Line 157: (prevalence?). Please delete or correct

Discussion:

Overall, this study has public health implications and it adds to the body of evidence that specific ethnic minority groups are at higher risk of 25OHD deficiency in Ireland. The authors should have a paragraph clearly describing the practical public health implications and future directions.

This study was performed pre-COVID, please indicate how you think your findings may or may not be different now

Limitations of the study: please expand this section as per above. Confounding factors should all be addressed in an attempt to explain possible reasons for 25OHD deficiency.

Lines 160 – 176: there are a few papers now addressing reasons for not taking vitamin D supplementation (UK study). O’Connor et al. 2018. Knowledge, Attitudes and Perceptions towards Vitamin D in a UK Adult Population: A Cross-Sectional Study. Int J Environ Res Public Health. 2018 Nov; 15(11): 2387.

Lines 160 – 176: the results from this study regarding the lack of differences seen between high vs. low synthesising periods is in line with reports from people diagnosed with chronic conditions. Please look add this to this section. Furthermore, there is evidence reporting that people from South Asia (residents in Ireland) do not tend to go on holidays and expose their skin to the sun as much as Caucasians due to cultural and religion reasons. Please discuss.

Lines 174 – 175: “There is no firm hypothesis to account for this difference – it is possible the different fat mass and muscle composition may have an effect while differences in the metabolic utilization or tissue priority for vitamin D could be different in the sexes”. This point should be explained further. There is evidence that higher physical activity levels and/or exercise release 25OHD from fat tissue (and maybe muscle tissue) into the circulation increasing plasma 25OHD. Although, physical activity levels/exercise was not measured here, boys and men tend to be more active than girls and women. I would suggest you look at articles on this topic published by Prof Dylan Thompson, University of Bath. 

Reviewer 2 Report

Many thanks for  your contributions.

However, vitamin d deficiency in minor population in Irish area as well as other Asian populations

is very common due to various factors.

In discussion area, to mention the immunity in relation of COVD-19 pandemci is not proper in this

cross-sectional results.

I wonder there is any difference of the vitamin d deficiency in the Irish South East Asian population

compare to other Asian population groups. Isn't the similar pattern in other Asian population due to

lack of sun-exposure, relatively dark skin color, poor vitamin d containing food, and etc.

I think to recommed to analysis the difference or other factors to affect the serum D level in study

population compare to native Irish population

Round 2

Reviewer 1 Report

Nutrients_988129 review_2

Title: A high prevalence of vitamin D deficiency observed in an Irish South East Asian population: A cross-sectional study

Authors: Eamon Laird, J-Bernard Walsh, Susan Lanham-New, Maria O’Sullivan, Rose Anne Kenny, Helena Scully, Vivion E. Crowley and Martin Healy

The authors have addressed some of the comments suggested and clarify many of queries. Please see further suggestions following the clarifications made by the authors in their response letter.

Further comments to the authors:

Line 48 – 54: the authors should be more specific when describing the different ethnicity populations.

Response: Although, there are cited references, it is of upmost importance to add the ethnic groups (in a standard manner) in the characteristics of the population as this can help guide future public health recommendations and studies. Furthermore, I suggest the authors present the 25OHD deficiency and insufficiency from these groups to highlight any potential differences. There is always ways in which the data can be stratified into smaller informative groups.

(Line 55: “The aims of this study, therefore, were to assess the vitamin D status for a selection of the Dublin (53.3°N; Capitol City) population of South East Asian descent and to provide baseline pilot data for this population”. I suggest that the authors explain what they mean by “baseline pilot data”. Is this an intervention study and the authors are presenting the baseline results? Is this preliminary data from a larger study? Response: We have now changed the phrase to pilot data. It is not an intervention – this is a cross-sectional observational study designed to provide pilot data on the vitamin D status of Irish South East Asians in order to be able to conduct a larger and broader study into this population. The data generated might, however, inform a future strategy to address the vitamin D status of this population)

Response: The above is clear now.

The authors should follow the STROBE checklist for the description of this article (introduction, methods, results and discussion). Response: The authors can assure the reviewer we have followed the STROBE checklist throughout whilst also following the layout as required by the journal.

Response: Please add this to the methods – a line indicating that you follow the STROBE checklist with an appropriate citation/reference should be sufficient and attach the strobe checklist as supplementary material.

The authors have collected the necessary information to report 25OHD status; however, there is key information missing here. For example, factors that may contribute to 25OHD deficiency and insufficiency apart from age and gender. For instance, ethnicity should be more specific, clinical history and any medications, socio-economic status, body mass index, vitamin D supplement usage, work and work patterns. Other information such as dietary intake (or cooking traditions), exposure to sunlight and time spent indoors, skin pigmentation and/or physical activity levels if possible. The authors should explain the rationale for choosing age and gender as main independent variables and for stratification purposes. Response: The authors are aware that this information is missing and have stated as such in the limitation section. This information was not collected as it was a convenient sample from GP results that arrived for analysis at the St James’s Hospital Biochemistry laboratory. The purpose of this paper was to highlight the low vitamin D status of this population in order to be able to conduct a more in-depth study that will include all the relevant information as stated by the author. However, in order for this to occur, initial pilot information is needed as this is the first time this population have been examined for vitamin D in an Irish context. Age and gender were selected as these were the other variables available with the data and are also two of the most important in the context of vitamin D as shown by this recent paper which was published in Nutrients and sets a precedent for this kind of analysis : ‘Scully H, Laird E, Healy M, Walsh JB, Crowley V, McCarroll K. Geomapping Vitamin D Status in a Large City and Surrounding Population-Exploring the Impact of Location and Demographics. Nutrients. 2020 Sep;12(9):2663’.

Response: I thank the authors for highlighting this interesting paper; however, if this data was obtained from GP practices, there will be other relevant information that may impact on vitamin D levels and therefore this should have been collected where possible. This is particularly important for a pilot study. Furthermore, it is well known that Asians and other members of the BAME community living in higher latitudes have lower vitamin D levels than Caucasians.

Line 59 Study design: please state what type of study design is your study. Response: We have added this in on line 57. We have also added this to the title.

Response: the study does make a lot more sense now.

Lines 75 – 90: please describe all the reference ranges used for your study including PTH. Did the authors collect phosphate? This is also an important marker of bone health.Response: We have added in the standard normal reference ranges for PTH and calcium as used by the hospital laboratory below Table 1 and Table 2. Unfortunately, as stated earlier, information such as phosphate was not available, but we would like to investigate this with a larger more in-depth study.

Response: thank you for clarifying

As per STROBE, please add a flow diagram. Response: The authors feel this would be inappropriate and not actually add any valuable information for the reader as we had actually used all the participants we had initially selected (none were <2 years, had missing data or were from Nursing homes). The authors felt it was important for the reader to see the exclusion criteria we would have used if it was needed.

Response: it is really useful from an epidemiological perspective to describe the total number of Asians (non-Chinese) registered, then the number of people that had vitamin D levels available and the number of people that did not. Furthermore, I suggest the authors report the total population of Asians (non-Chinese) living in your catching area. This will allow to perform a power calculation to be able to include a representative sample in your future research.

Interquartile range (IQR) should accompanied median values. Please add this where relevant throughout the manuscript. Response: We have added the 25-75th percentile with the median values in the manuscript where appropriate and only when they have not been reported in Tables. 12). All p values should be low case and italic. Response: All p values have been converted to lower case and italic. 13). Line 99: indicate the total sample size. Response: We have indicated this on line 101. 14). Line 110: “were vitamin D deficient, with 16.7% sufficient”. Please delete with.Response: This has been deleted as requested.

Response: great. Thank you.

Lines 119 – 121: the authors should stratify the data by age and look at whether there is any correlation between 25OHD and PTH in children. Response: The authors have examined the 25(OH)D differences by age in the results and also in Figures 3 and 4. Given the small number of those <18 this analysis would not be feasible while the relationship of PTH with 25(OH)D in this age group was not the focus or hypothesis of this analysis although it is something we would like to examine in future studies.

Response: thank you for the clarification re-small sample size (<18 years). Since this is not representative of the larger sample and it might provide an inaccurate vitamin D status representation of children and adolescents and young adults, I suggest the authors remove this from the analysis and include only adults. Furthermore, with the data that you have it is appropriate to perform correlation analysis including age and vitamin D levels or associations (chi-sq test) looking at associations between age groups and vitamin D status (the same for gender). If power calculation supports, the authors could also perform univariate analysis.

“We had to use a different format as one set of graphs represent gender while the other is age groups where adding the numerous p values to the age group would make it difficult to understand. The comparison values have been added to the results section. Value were added to the top of the columns in order for the reader to easily see the value without guessing from the axis whilst this is typically done in Nutrients. We did not add the n as these are given both in the tables and main text as it can be easily worked out/seen by the reader and it is not good practice to repeat values that are already presented elsewhere in a paper. By adding the n we could confuse the reader as would we add the n for each age group or the n for each deficiency group? We did not want to confuse the reader”.

Response: I still suggest these are added to the graphs as most of the groups are below 100. This can easily be done without confusing the reader and it is common practice. They can be added below each column as labels.

21). Overall, this study has public health implications and it adds to the body of evidence that specific ethnic minority groups are at higher risk of 25OHD deficiency in Ireland. The authors should have a paragraph clearly describing the practical public health implications and future directions. Response: The authors agree with the reviewer and there is a section before the limitations where we discuss this ‘Recently the Food Safety Authority of Ireland (FSAI) updated the national vitamin D intake guidelines for the Irish population as a whole. [31]. In the policy document it was recommended that food (including supplements) need to provide 10 ug of vitamin D every day for everyone aged ≥5 years. Few foods, however, contain the necessary vitamin D concentrations and those that do are infrequently consumed. This may be due to lack of availability or costs of the food products. There may also be cultural traditions regarding different food types including, in particular, vegan diets. Importantly in the current context, although other international policy documents address the vitamin D needs of specific population sub-sets, the FSAI does not do so. However, across the EU there is little harmonisation of vitamin D supplementation and fortification policies and mandatory food fortification with vitamin D is rarely implemented. Also, policy documents can include confusing and contradictory information which makes it difficult to translate guidelines to specific settings [32]. In the UK both the Scientific Advisory Committee on Nutrition (SACN) and the National Institute for Health and Care Excellence (NICE) have made recommendations for ‘at risk’ groups regarding supplementation with vitamin D [33,34]. These risk groups include those on vegetarian diets, have dark skin pigmentation, have limited exposure to UVB light because of cultural or lifestyle practices and would include those in the Asian community. A ‘multi-agency’ approach is recommended to improve communication and education relating to vitamin D intake and encourage behavioral involvement in uptake of recommendations’.

Response: thank you for clarifying this and apologies for not being more specific in what I meant. I agree this should remain and that it talks about public health implications; however, in my opinion the authors should be more specific i.e. public health implications that specifically tackle this group and/or highlight what is done regarding vitamin D in South-East Asia. What do the authors suggest that could be done to reach this population? NICE/SACN recommendations are clearly not reaching these populations and this why I suggested the paper by O’Connor, because it provides an overview of knowledge, attitudes and perceptions on vitamin D intake, which as you suggested in your introduction and discussion is of paramount importance to achieve appropriate levels.

Another point: Could the authors please add a section in the discussion explaining how future research should be performed? For instance, specific aspects that need to be researched, a power calculation indicating the sample size and importantly all relevant variables that should be included. This can be presented in a “box”.